# Biomarkers-in-Cardiology 8 RE-VISITED—Consistent Safety of Early Discharge with a Dual Marker Strategy Combining a Normal hs-cTnT with a Normal Copeptin in Low-to-Intermediate Risk Patients with Suspected Acute Coronary Syndrome—A Secondary Analysis of the Randomized Biomarkers-in-Cardiology 8 Trial

**DOI:** 10.3390/cells11020211

**Published:** 2022-01-08

**Authors:** Evangelos Giannitsis, Tania Garfias-Veitl, Anna Slagman, Julia Searle, Christian Müller, Stefan Blankenberg, Stephan von Haehling, Hugo A. Katus, Christian W. Hamm, Kurt Huber, Jörn O. Vollert, Martin Möckel

**Affiliations:** 1Medizinische Klinik III, Department of Cardiology, Angiology and Pulmology, University Hospital of Heidelberg, 69120 Heidelberg, Germany; hugo.katus@med.uni-heidelberg.de; 2Department of Cardiology and Pneumology, University of Göttingen Medical Center, German Center for Cardiovascular Research (DZHK), Partner Site Göttingen, 37075 Göttingen, Germany; tania.garfiasveitl@med.uni-goettingen.de (T.G.-V.); stephan.von.haehling@med.uni-goettingen.de (S.v.H.); 3Department of Emergency Medicine Campus Mitte and Virchow and Department of Cardiology, Charité-Universitätsmedizin, 13353 Berlin, Germany; Anna.Slagman@charite.de (A.S.); Julia.Searle@baek.de (J.S.); martin.moeckel@charite.de (M.M.); 4Cardiovascular Research Institute Basel (CRIB) and Department of Cardiology, University Hospital Basel, University of Basel, 4031 Basel, Switzerland; Christian.Mueller@usb.ch; 5Department of Cardiology, University Heart and Vascular Centre Hamburg, 20251 Hamburg, Germany; s.blankenberg@uke.de; 6Department of Cardiology, Campus Kerckhoff, University of Giessen, 61231 Bad Nauheim, Germany; C.Hamm@kerckhoff-klinik.de; 73rd Department of Internal Medicine, Cardiology and Intensive Care Medicine, Wilhelminenhospital, 1060 Vienna, Austria; kurt.huber@meduniwien.ac.at; 8BRAHMS GmbH Deutschland, 16761 Berlin, Germany; joern.vollert@thermofisher.com

**Keywords:** acute coronary syndrome, randomized trial, risk stratification, safe discharge, copeptin, high sensitivity troponin T

## Abstract

Regarding the management of suspected Non-ST-segment-elevation acute coronary syndrome (ACS), the main Biomarker-in-Cardiology (BIC)-8 randomized controlled trial study had reported non-inferiority for the incidence of major adverse cardiac events at 30 days in the Copeptin group (dual marker strategy of copeptin and hs-cTnT at presentation) compared to the standard process (serial hs-cTnT testing). However, in 349 (38.7%) of the 902 patients, high-sensitivity cardiac troponin was not available for the treating physicians. High sensitivity cardiac troponin T was re-measured from thawed blood samples collected at baseline. This cohort qualified for a re-analysis of the 30-day incidence rate of MACE (death, survived cardiac death, acute myocardial infarction, re-hospitalization for acute coronary syndrome, acute unplanned percutaneous coronary intervention, coronary bypass grafting, or documented life-threatening arrhythmias), or components of the primary endpoint including death or death/MI. After re-measurement of troponin and exclusion of 9 patients with insufficient blood sample volume, 893 patients qualified for re-analysis. A total of 57 cases were detected with high sensitivity cardiac troponin T ≥ 14 ng/L who had been classified as “troponin negative” based on a conventional cardiac troponin T or I < 99th percentile upper limit of normal. Major adverse cardiac events rates after exclusion were non-inferior in the Copeptin group compared to the standard group (4.34% (95% confidence intervals 2.60–6.78%) vs. 4.27% (2.55–6.66%)). Rates were 53% lower in the per-protocol analysis (HR 0.47, 95% CI: 0.18–1.15, *p* = 0.09). No deaths occurred within 30 days in the discharged low risk patients of the Copeptin group. Copeptin combined with high sensitivity cardiac troponin is useful for risk stratification and allows early discharge of low-to-intermediate risk patients with suspected acute coronary syndrome is as safe as a re-testing strategy at 3 h or later.

## 1. Introduction

Recently, the 2020 ESC Guidelines on non-ST-segment elevation acute coronary syndrome (NSTE)-ACS [1] were published and now endorse the use of high sensitivity cardiac troponin (hs-cTn) assays and fast diagnostic protocols. For several debatable reasons [2], the dual marker strategy (DMS) that allows an instant rule-out of myocardial infarction (MI) when copeptin and cTn are below respective cut-offs was discouraged despite the recommendation [1] to consider copeptin whenever a high sensitivity cTn assay is not available.

The recommendation to use copeptin together with a conventional and contemporary sensitive cTn assay (cTn) has beneficial impact on the daily routine in ruling out NSTE-ACS since these assays are still used in the majority of hospitals worldwide [3]. But despite the new 2020 ESC Guidelines on NSTE-ACS [1], the vast majority of observational trials found significantly improved sensitivities and negative predictive values, even if copeptin was combined with a hs-cTn assay [4,5,6,7]. Inconsistent findings on the added diagnostic value over hs-cTn were seen whenever the analysis was not limited to the comparison of sensitivities and negative predictive values (NPVs) but across the entire diagnostic spectrum, including rule-in [7,8]. Moreover, inferior prognostic performance of DMS was observed in studies where retrospective analysis was not restricted to patients with low-to-intermediate risk [9,10].

Secondly and more importantly, the safety of discharge using fast hs-cTn-based protocols has mainly been evaluated retrospectively from observational trials where physicians were blinded to assays and protocols and where management decisions were left at the discretion of the treating physicians [11,12,13]. In retrospective analyses, mortality rates within 30 days after discharge were consistently very low [14]. So far, data on the prognostic performance and on length of emergency department (ED) stay and discharge rates are limited to two prospective observational registries [15,16] and a single randomized trial [17]. The Biomarker-in-Cardiology (BIC) 8 trial [18], a large interventional multicenter trial that randomized low-to-intermediate risk patients to early discharge or standard strategy using a single biomarker combination of Copeptin combined with cardiac troponin was not considered relevant for the current guideline recommendations, referring to the limitation that an hs-cTn assay was used in only 54% of the entire study cohort.

Copeptin (CT-pro-vasopressin) is a marker of acute (haemodynamic) stress [19,20] and is elevated immediately at the presentation of patients with acute MI [21]. Conversely, serving as a proof of principle, copeptin levels remain below the respective rule-out cutoff in patients with unstable angina [21]. A pooled analysis of randomized and observational studies have shown that due to complementary pathophysiology and release kinetics, copeptin in combination with conventional and with high sensitivity cardiac troponin is associated with an NPV of up to 99% or higher [21,22,23].

In the light that 2020 ESC Guidelines [1] have now also shifted patients from the former intermediate-risk group that comprised patients with normal cardiac troponin but previous percutaneous coronary intervention (PCI) or coronary artery bypass grafting (CABG), previous MI, diabetes mellitus, chronic kidney disease to the low-risk group, the need to improve risk stratification in this group and also to guide the safety of early discharge and further outpatient management has increased. Although DMS seems attractive for management of low-to-intermediate risk patients with suspected ACS where an RCT backs earlier discharge from ED without an excess risk compared to a standard serial troponin-based protocol, the usefulness of DMS has been questioned if a hs-cTn assay is routinely available.

Therefore, the working hypothesis of this substudy was to test whether copeptin retains its ability to risk stratify and guide early discharge regardless of whether copeptin is combined with cTn or hs-cTn. For this secondary analysis of the BIC-8 trial, all cTn values were replaced by hs-cTnT values that were measured from stored frozen blood samples.

## 2. Materials and Methods

The BIC-8 trial [18] was a multicenter, interventional clinical process randomized controlled trial (RCT). Briefly, participants were recruited in the EDs and/or chest pain units (CPUs) in five German, one Swiss, and one Austrian site from April 2011 until May 2013. Patients were eligible if they were aged ≥18 years, presented with signs and symptoms of ACS, and had negative troponin values, defined as hs-cTnT or cTnI < 99th percentile or a conventional cTnT (cTnT) (Radiometer Deutschland, Krefeld; 30 ng/L ULN) at presentation. Patients were excluded if they were diagnosed with ST-elevation myocardial infarction (STEMI), if hospital admission was indicated due to high risk as defined in current guidelines (continuing chest pain or recurrent episodes of chest pain under therapy, GRACE score above 140), and if hospital admission was necessary for any other reason. Patients were then randomized using 1:1 computer-generated block randomization into a Copeptin group and a standard group where copeptin results were not revealed to the treating staff. In the copeptin-guided arm, cases with a negative result for copeptin (<10 pmol/L) and a cardiac Troponin < cutoff were considered as low risk and could, per protocol, be discharged into ambulatory care. However, the final decision to admit or discharge was left at the discretion of the treating physician. In some cases, physicians overruled the per-protocol decision for discharge and managed patients differently and at their discretion. These patients were labeled as protocol deviations.

The BIC-8 study complied with the Declaration of Helsinki and received ethics approvals from all study sites’ ethics committees. All patients provided written informed consent. The study is registered at the German Clinical Trials Register (DRKS00000276), the International Clinical Trials Registration platform of the WHO (UTN U1111-1118-1665), and at ClinicalTrials.gov (NCT0149873).

### 2.1. Biomarker Testing

In the BIC-8 trial, cardiac troponin was tested by routine practice at the individual sites. A cTnT point-of-care-troponin (POCT) assay (AQT 90 -Radiometer Deutschland, Krefeld) was used at two sites (cut-off ≥ 30 ng/L). Hs-cTnT (Roche Ltd., Rotkreuz, Switzerland) was used at four sites for initial and serial measurements, and at two sites for serial measurements but not for the baseline measurement at admission. One site (KH) used conventional troponin I (Siemens Dimension-System, Siemens Healthcare Diagnostics Inc, Tarrytown, NY, USA) (cut-off level 56 ng/L) until November 2012 and troponin I (Siemens Dimension Vista-System) thereafter (cut-off level 45 ng/L). Overall, cTnT was tested in 349 (38.7%) cases, hs-cTnT in 487 (54.0%), and cTnI in 66 (7.3%) cases at presentation. For subsequent measurements, hs-cTnT was measured in all but the 66 cases where cTnI was routinely used. In all cases without hs-cTnT value at baseline, concentrations were re-measured batchwise from previously unthawed aliquots that had been stored at −80 °C, using the Roche hs-cTnT assay on a Cobas 411. A hs-cTnT could not be measured in 9 cases due to missing samples. Using Cobas411, the limit of blank (LoB) was 3 ng/L, and the limit of detection was 5 ng/L. The 99th percentile of a healthy reference population is 14 ng/L, with a coefficient of variation of 10% at 13 ng/L [24].

Copeptin was measured from the routine blood sample at admission, using the Thermo Scientific B.R.A.H.M.S Copeptin ultrasensitive Kryptor assay (BRAHMS Thermo Fischer Deutschland, Berlin, Germany). The assay has a detection limit of 0.9 pmol/L and a functional assay sensitivity of <2 pmol/L. For the rapid rule-out algorithm, the cutoff value was 10 pmol/L. The cut-off was chosen with reference to Keller et al. [25], who determined different cut-offs in a reference population and tested the diagnostic performance of these cut-offs in an ACS population.

### 2.2. Patient Exclusion after Retesting of hs-cTnT Instead of cTnT or cTnI

Patients with a hs-cTnT ≥14 ng/L would not have qualified for inclusion in BIC-8 and were excluded for BIC-8 RE-VISITED.

### 2.3. Outcomes

The primary endpoint was the proportion of combined major adverse cardiac event (MACE), defined as all-cause death or survived sudden cardiac death, acute MI, re-hospitalization for ACS, acute unplanned PCI, CABG, and documented life-threatening arrhythmias (ventricular tachycardia, ventricular fibrillation, atrioventricular-block III) within 30 days, including events during the index hospital stay. Acute MI was defined as per the 3rd Universal Definition of MI, which was actual at the time of the study. Every patient was assigned only one MACE with priority for the event that occurred first.

For the analysis of BIC-8 RE-VISITED, other secondary endpoints, including proportion of coronary angiography, bleeding events per Thrombolysis-in-Myocardial-Infarction (TIMI) definition, discharge rates, and length of stay were not evaluated given that the present study represents a post-randomization cohort.

### 2.4. Statistical Evaluation

Statistical testing of categorical variables was performed using exact binomial tests. For numerical variables, t-tests (in case of normal distribution) or Wilcoxon tests (no normal distribution) were employed. Comparison of two groups was performed using the Chi-square test or Fisher’s exact test. Data are provided as means with standard deviation in case of a normal distribution, or otherwise as medians with 25th and 75th percentiles. Discrete variables are reported as absolute values, with relative frequencies in brackets.

Basic assumptions on sample size and on the calculation of the non-inferiority margin were reported earlier [18]. Briefly, a total number of 446 participants per group (892 overall) were calculated as appropriate to the sample size, with a power of 80% and a level of significance of 5%. Confidence intervals (CI) for differences were calculated as one-sided 97.5% CI using the Wilson procedure with a correction for continuity. The non-inferiority margin was set at 5%.

Given that the present study represents a post-randomization analysis, statistical evaluations were restricted to the intention-to-treat and the per-protocol analysis. Other analyses on secondary endpoints were not carried out.

A *p*-value below 0.05 was considered significant. All tests were performed using the software packages SPSS (IBM SPSS Statistics, Version 26, Armonk, NY, USA) and SAS Institute, Version 9.3, Cary, NC, USA.

## 3. Results

### 3.1. Patient Characteristics

A total of 902 patients were enrolled in the BIC-8 trial. Of these, blood specimens for measurement of hs-cTnT were not available in 9 patients. Another 57 patients with a normal cTnT or cTnI on admission were found to have a hs-cTnT ≥ 14 ng/L after re-measurement from stored blood material and were excluded from the present substudy. Accordingly, the cohort evaluated in BIC-8 RE-VISITED consisted of 836 patients (Appendix A; consort diagram). The baseline characteristics of the standard and the Copeptin group are displayed in Table 1. The time from chest pain onset to the first blood draw was 3.39 ± 2.83 h. The time from admission to the first blood draw was a mean of 0.08 ± 0.56 h.

### 3.2. Serial Troponin Protocol in the Overall Cohort and in the Standard of Care

Among the 836 patients who qualified for BIC-8 RE-VISITED, cardiac troponin (either cTnT, hs-cTnT or cTnI) was available in all subjects at baseline and was re-tested in 384 cases after 3 h, in 218 cases after 6 h, and in 156 cases after 3 and 6 h. In the copeptin arm, 343 of 414 patients (82.9%) qualified for DMS. (Appendix A). In the standard-of-care, the 0/3 h protocol was used in 77.5% (327 of 422 patients).

### 3.3. Effect of Exclusion on Corresponding MACE

A total of 57 cases were excluded. The distribution of MACE by groups and by hospital discharge in the BIC-8 RE-VISITED study cohort is displayed in Figure 1.

### 3.4. Protocol Deviations and Analyses per Protocol

In the standard group, a protocol deviation was registered in 32 cases (7.6%) because a second hs-cTnT was missing in 32 cases. In the Copeptin group, a protocol deviation was seen in 74 cases (17.9%). Of these, 6 cases had a copeptin ≥ cutoff, and 68 cases had copeptin < 10 pmol/L.

Among cases with a copeptin ≥ cutoff, a second hs-cTnT was missing in 4 cases, and 1 case was enrolled beyond the 12-h interval from the last chest pain episode. Another patient with elevated copeptin was diagnosed with an acute aortic syndrome.

Among the 68 patients with copeptin < cutoff, 67 cases were admitted to hospital instead of being discharged (overruled), and 1 patient left hospital against the recommendation for hospital admission.

### 3.5. Outcomes within 30 Days

Event rates of the primary endpoint and its components in the groups are displayed in Table 2 and Appendix A (Appendix A). Hazard ratios (HR) for MACE at 30 days (Figure 2) were similar in the randomized groups compared to the original BIC-8 trial for the intention-to-treat analysis that included per-protocol and protocol deviations. Using the same non-inferiority margin, the absolute risk difference between the standard group and copeptin group was comparable, confirming the non-inferiority of DMS strategy using copeptin with hs-cTnT versus standard strategy (Figure 1). The per-protocol analysis, i.e., adherence to the proposed discharge of low-risk patients, demonstrated a higher risk reduction for MACE favoring the Copeptin group with an HR of 0.47 that showed a trend for statistical significance suggesting higher benefits with protocol adherence and thus corroborating the usefulness of DMS with either cTn or hs-cTnT for risk stratification and guidance of safe early discharge.

Detailed information on all copeptin-negative patients with MACE is provided in the Appendix A.

MACE rates in randomized groups and by protocol adherence are displayed in Table 2. Briefly, irrespective of randomization, MACE rates in patients discharged home versus admitted were 0.9 versus 18.8% (6 of 676 cases vs. 30 of 160 cases, *p* < 0.001).

### 3.6. Disposition of Patients in Randomized Treatment Arms and Associated MACE Rates at 30 Days

A total of 675 patients were discharged and 161 cases were admitted. A display of patient disposition in randomized arms and associated MACE rates at 30 days is shown in Figure 2.

In the copeptin arm, 327 of 414 patients were discharged, and 87 cases were admitted. Among those discharged, 284 cases were discharged directly from ED, whereas 43 cases were discharged after admission to the CPU.

Of the standard arm, 348 of 422 cases were discharged and 74 were admitted. Among those discharged, 51 were discharged directly from ED, whereas 297 cases were admitted to the CPU before discharge.

Overall MACE rates in the copeptin arm after discharge were 0.6% in the entire discharge group (2 of 327 cases) and 0.7% (2 of 284 cases) in cases discharged directly from the ED. In the standard arm, MACE rates after discharge were 1.1% (4 of 348 cases) in the overall discharge group and 0% (none of 51 cases) in cases discharged directly from the ED. Overall, MACE rates were lower among those discharged primarily from the ED than those discharged home after an interim observation in the CPU.

Among admitted patients, a total of 30 events occurred, a bit more than a third (40%, *n* = 12) in those with protocol deviations.

We refrained from a secondary analysis of the secondary endpoints (decision for coronary angiography, hospital discharge, bleeding rates, length of ED stay) as all management decisions depended entirely on the treating physicians who were unaware of re-measured hs-cTnT results.

## 4. Discussion

This secondary analysis of the randomized BIC-8 trial [18] reports three major findings. First, the performance of DMS for the guidance of early discharge following instant rule-out of MI is consistently non-inferior to a standard serial sampling protocol. In the BIC-8 trial, the standard protocol consisted of the ESC 0/3 h protocol in 77% of cases. This finding corroborates findings from the BIC-8 trial where a conventional or contemporary sensitive cTn assay had been used for the initial blood testing in almost 50% of the entire study population. The MACE rates would have been even lower if patients had been managed per protocol. Our findings, together with confirmatory results from a multicenter observational study [25], should increase confidence for the safety of discharge and stimulate higher adherence to DMS-guided management. Seemingly, the use of a hs-cTnT instead of cTn assay improved risk stratification in both groups, a finding that has previously been confirmed in many trials that evaluated the effect of a hs-cTn assay instead of a conventional cTn on mortality [26,27,28], and is attributed to the identification of small albeit prognostically relevant myocardial injury. Consistently, accumulating evidence suggests that copeptin provides additional prognostic information on the risk of death that are independent of the information provided by hs-cTn or clinical scores such as the GRACE score. In particular, an elevated copeptin suggests the presence or co-existence of a prognostically relevant acute disease and thereby contributes to the interpretation of differential diagnoses, particularly when an MI has been excluded. As such, it is tempting to speculate that combination of copeptin with hs-cTn would allow an RCT-based guidance of safe discharge on the one hand, and an improvement of risk stratification by taking advantage of the superior role of hs-cTn instead of cTn, on the other hand.

Second, as a comparator to the copeptin arm, the 0/3 h diagnostic protocol was used in two-thirds of patients assigned to the standard-of-care, which was already noteworthy at the time when BIC-8 was performed. We presume that higher utilization of the 0/6 h protocol that was widely used at the time of study conduct would have increased the difference in length of ED stay. However, the high utilization rate of the ESC 0/3 h protocol in the BIC-8 trial gives the opportunity to compare the DMS strategy with the serial cTn testing strategy that is—until today—employed in the majority of sites worldwide, when a validated hs-cTn assay is routinely available. Noteworthy, at the time when BIC-8 trial was designed, the ESC 0/1 h protocol was not established. Still, a global survey on utilization rates of hs-cTn assays and fast protocols [3] reported that fast protocols, in particular the ESC 0/1 h algorithm, are not in widespread use, although this may have changed since 2017 when the survey was conducted.

Third, our data confirm that low-to-intermediate risk patients may be discharged home safety after rule-out, with a comprehensive MACE rate at 30 days below 1%, unless the default decision to discharge is over-ruled based on clinical judgement beyond screening for eligibility based on clinical scores. However, the disproportionate clustering of events in patients (*n* = 12) where the discharge recommendation was overruled (a third of all events in a group that comprised only 12.5% of the entire study cohort) suggests that these patients were either correctly identified by the treating physician as not being at low risk, thus underscoring the importance of clinical judgment in the overall process of decision making in the ED. Conversely, the findings suggest that low-risk patients might be exposed to a risk of avoidable events due to unnecessary coronary angiographies and/or pharmacological interventions. However, the exact reasons remain highly speculative, and neither the BIC-8 main study nor the present substudy allows any conclusions.

Our findings from BIC-8 RE-VISITED are very consistent with previous findings from the ProCore multicenter observational study [25] that demonstrated a reduction of ED stay and low mortality rates after primary discharge using DMS

This secondary analysis of the randomized BIC-8 trial indicates that a DMS where a normal copeptin is combined with a normal hs-cTnT is associated with a low risk for MACE after discharge and is as safe as a strategy that is based on serial re-testing of hs-cTnT.

### Limitations

Neither the BIC-8 trial nor the secondary analysis evaluated DMS against the ESC 0/1 h algorithm that is now being recommended as the preferred strategy in the current 2020 ESC Guidelines [1]. However, a survey on the worldwide use of hs-cTn assays and fast protocols has revealed the substantial, very limited implementation of the ESC 0/1 h algorithm, even in centers where hs-cTn assays were routinely available [3].

Our study results are confirmatory and have to be interpreted with scrutiny given that a post-randomization bias cannot be excluded. However, our findings are in line with established evidence that the use of hs-cTn instead of cTn improves risk stratification since more sensitive assays allow for the detection of prognostically adverse myocardial injury that would have been undetected with less sensitive assay generations [27,28,29]. Our findings are consistent with other observational studies that calculated future risk for MACE post-discharge based on risk modeling using different biomarker thresholds for hs-cTn [11,12,13]. The vast majority of these trials managed patients according to standard processes and not based on the optimal decision thresholds for rule-out or rule-in.

Finally, the original study was performed in the EDs under routine conditions and decisions on management that were made by treating ED physicians. Thus, inclusion of certain patient groups including older individuals, patients with advanced CKD, or those presenting with dyspnea or atypical symptoms as the leading symptom, could have given different outcome results in clinical routine. However, in order to confirm the original study findings of the RCT, a multicenter, international, real-world observational study [26], representing regional and geographic differences as well as including academic and non-academic hospitals, consistently confirmed the BIC-8 data and re-assure the safety of DMS with hs-cTnT.

## Figures and Tables

**Figure 1 cells-11-00211-f001:**
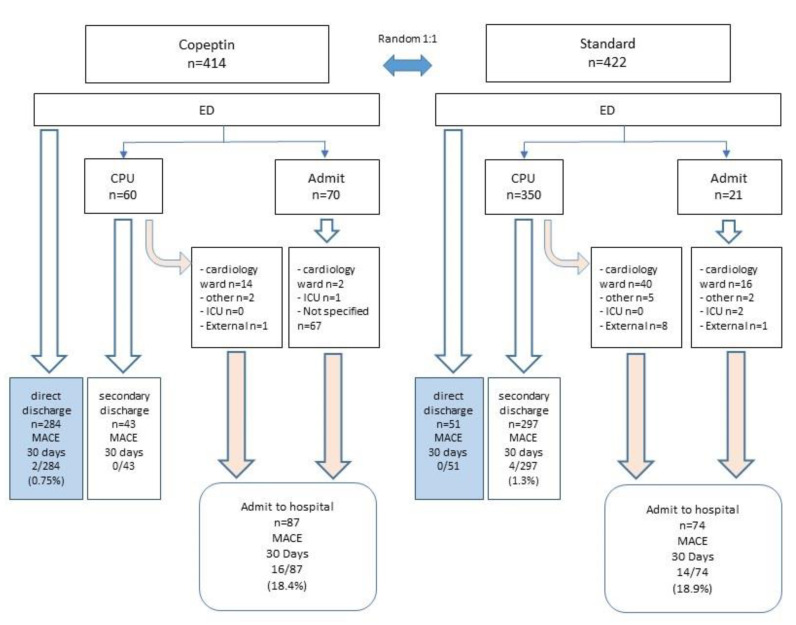
BIC-8 RE-VISITED. Patient disposition in randomized groups and MACE rates at 30 days. The numbers of cases in the corresponding boxes are displayed as absolute numbers and relative frequencies in brackets.

**Figure 2 cells-11-00211-f002:**
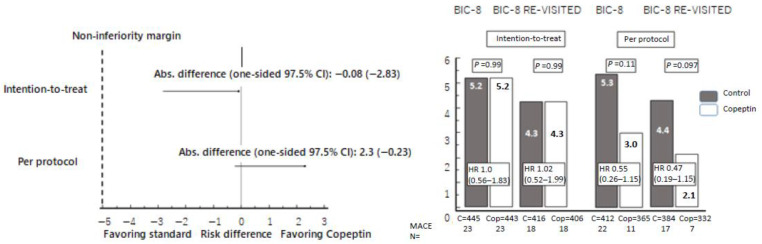
Left panel: Forest plot for differences in major adverse cardiac event (MACE) proportions. Absolute differences in MACE proportions within 30 days between the study groups with one-sided 97.5% CIs. The non-inferiority margin was prospectively defined at 5%. In none of the performed analyses, the non-inferiority margin was exceeded. Right panel: Proportions of MACE and corresponding hazard ratios (HR) with 95% confidence intervals (95% CI) according to intention-to-treat (including per protocol and protocol deviations), or per protocol treatment, both in the original BIC-8 trial and in the secondary analysis (BIC-8 RE-VISITED).

**Table 1 cells-11-00211-t001:** Characteristics of all patients and in the groups.

	All Patients (*n* = 836)	Standard Group (*n* = 422)	Copeptin Group (*n* = 414)
Patients’ characteristics			
Age (years) (mean ± SD)	53.3 ± 15.4	53.2 ± 14.9	53.4 ± 16.0
Male sex	62.0 (518)	64.2 (271)	59.7 (247)
Risk factors			
BMI	27.3 ± 4.79	27.3 ± 4.56	27.2 ± 5.00
Diabetes	12.9 (107)	13.2 (55)	12.6 (52)
Hypertension	56.8 (469)	56.7 (236)	56.8 (233)
Hyperlipidaemia	43.6 (354)	45.1 (184)	42.1 (170)
Family history of MI	27.4 (214)	24.5 (95)	30.4 (119) **
Smoker	33.2 (270)	35.1 (144)	31.2 (126)
Ex-smoker	31.1 (253)	29.8 (122)	32.4 (131)
GRACE-score (mean ± SD)	78.76 ± 26.7	78.12 ± 26.4	79.42 ± 27.1
TIMI risk score (Median/IQR)	1 (0–2)	1 (0–2)	1 (0–2)
Medical history			
Known CAD	25.2 (207)	24.5 (101)	26.0 (106)
Prior MI	13.3 (110)	14.4 (60)	12.2 (50)
Prior PCI	21.3 (176)	21.3 (88)	21.4 (88)
Prior CABG	4.3 (36)	3.1 (13)	5.6 (23) **
Chronic heart failure	5.6 (46)	3.9 (16)	7.4 (30) *
Primary valve disease	7.0 (57)	7.1 (29)	6.9 (28)
Prior valve surgery	1.2 (10)	1.0 (4)	1.5 (6)
Cardiomyopathy	1.8 (15)	0.5 (2)	3.2 (13) *
Renal disease	4.9 (40)	3.6 (15)	6.1 (25)
Time since symptom onset			
0–3 h (less or equal 3)	35.2 (294)	36.7 (155)	33.6 (139)
<6 h	42.5 (355)	44.1 (186)	40.8 (169)
<12 h	52.3 (437)	51.9 (219)	52.7 (218)
unknown	170	79	91

BMI, body mass index; TIMI, thrombolysis in myocardial infarction; CAD, coronary artery disease; MI, myocardial infarction; PCI, percutaneous coronary intervention; CABG, coronary artery bypass graft. Continuous variables are given as means with standard deviation. Discrete variables are reported as relative frequencies with absolute numbers in brackets. * *p*-value < 0.05, ** *p*-value < 0.01.

**Table 2 cells-11-00211-t002:** Primary endpoint analyses.

	Standard Group (*n* = 422)	Copeptin Group (*n* = 414)	Absolute Differenc in MACE Proportion
**MACE at 30 days**						
Yes	18	18	
No	398	388	
Unknown (lost to FU)	6	8	
MACE % (95% CI)			
Intention to treat analysis	4.27 (2.55–6.66) (18/422)	4.34 (2.60–6.78) (18/414)	−0.08 (−2.83)
HR = 1.019 (95% CI: 0.523 to 1.987), *p* = 0.99 (chi^2^)			
Exclusions per protocol deviation	32	74	
**MACE after Exclusions**			
Yes	17	7	
No	367	325	
Unknown	6	8	
MACE % (95% CI)			
Per protocol analysis	4.35 (2.56–6.88) (17/390)	2.05 (0.83–4.20) (7/340)	2.30 (−0.23)
HR = 0.472 (95% CI: 0.193 to 1.153), *p* = 0.09 (chi^2^)			

Analysis of the primary endpoint: All MACE within 30 days. The CIs for the absolute difference between the proportions in the respective study groups did not exceed the 5% non-inferiority margin in any analysis, confirming non-inferiority of the copeptin-based process as hypothesized.

## Data Availability

The main study results have been published previously and are cited as reference #18.

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
