# Peer review of "Biomarkers-in-Cardiology 8 RE-VISITED—Consistent Safety of Early Discharge with a Dual Marker Strategy Combining a Normal hs-cTnT with a Normal Copeptin in Low-to-Intermediate Risk Patients with Suspected Acute Coronary Syndrome—A Secondary Analysis of the Randomized Biomarkers-in-Cardiology 8 Trial"

_cells, 2022, doi:10.3390/cells11020211_

Round 1

Reviewer 1 Report

This is a study about the application of copeptin in acute chest pain patients. In contrast to serial conventional cTn protocol, dual marker strategy allows an instant rule-out of myocardial infarction.

  1. DMS was proved in multiple studies previously and had incremental value. Although the samples originated from BIC-8 trial, the setting of the study may not be the same since the goal of the study is different. DMS was only proved for instant rule-out of myocardial infarction, it may be more appropriate to compare the patients discharged by DMS using copeptin and hs-cTnT and the 0/3 hour protocol using hs-cTnT. For this purpose, patients discharged from ED and CPU should be the target group rather than all the patients. An analysis of these patients divided into DMS (using hs-cTnT) and 0/3 hour protocol is suggested.
  2. The order of figures is not correct. Figure 2 appeared first.
  3. The abbreviations should be written in their full names when they were showed first time in the text, such as CPU.

Author Response

1. The present study is a substudy of BIC-8 and has exactly the same goals. However, in contrast to the original BIC-8, this substudy studied the effects of Copeptin combined with a hs-cTnT < 99th percentile URL in all patients. Therefore, a comparison with another (very interesting) goals, e.g. the comparison of DMS with ESC 0/3 hour protocol is not possible. 

2. We corrected the sequence of figures

3. We followed the suggestion of the reviewer and used abbreviations only upon the second appearance in the text.

Reviewer 2 Report

The authors did a good job demonstrating the importance of using copeptin with hs-cTnT in patients with low-to -intermediate risk of ACS. In my opinion, the authors should replace the figure 2 (unclear). 

Author Response

We thank the reviewer for his positive attitude and his Suggestion to omit figure 2.

We believe that Figure 2 is important to highlight the consistency of BIC-8 with BIC-8 Revisited, and thus indicating that DMS works as well with cTn as with hs-cTn. Another important aspect is that protocol adherence was associated with the largest reductions in events. Therefore, we would prefer to keep this figure in the main text.    

Reviewer 3 Report

In this manuscript, Giannitsis et al. aimed to re-evaluate the conclusion of BIC-8 RCT study by re-measuring the hs-cTnT from thawed blood samples. The 30-day incidence rate of MACE was evaluated for the secondary analysis. Although this was a follow up study with compromised novelty, the results may provide confirmatory evidence and guideline for dual marker strategy in the future management of NST-ACS.

This manuscript should be revised based on the following concerns:    

  • Figure quality: the order of Fig1 and Fig2 should be reversed based on the order of the results presented.
  • Figure 2 was in poor resolution and the unnecessary wavy line mark.
  • In page 7, section 3.6, paragraph 5, the author mention that “Among admitted patients, …, a bit more than a third (40%, n=12) in those with protocol deviations.” Here the n=12 was not clearly defined before. Please clarify in Fig2 (after revision, should be Fig1) and in the method section.
  • The author highlighted several times of “to manage per protocol”, please define this concept more clearly.
  • The author cited 29 reference in the main text, which seemed to be missed in the reference section.

Author Response

We thank the reviewer for his valuable and very constructive comments. Obviously, we did not explain important terminologies such as per protocol and intention-to-treat appropriately. We added text to explain the Terms and expand on anticipated effects of biomarker protocol adherence 

  1. We corrected the wrong sequence of figures 1 and 2
  2. We clarified the term protocol deviations and added text to discuss the disproportionate excess of events among patients where physicians overruled the protocol decision to discharge.
  3. In fact, we incorrectly used reference #26 instead of reference #25. Therefore, there is no reference #29 and the three cited articles are now referenced correctly as #26-28.

Reviewer 4 Report

Regarding the title, the abbreviation (ACS) can be removed - it is not necessary.

The title could state a major finding of the study.

The abstract (especially when read alone) may not be clear for those who are not familiar with previously published study (Biomarker-in-Cardiology 8 trial) - what is the copeptin group? standard process?

The last sentence of the abstract should be a strong summary statement of the study. I also suggest summarizing the main findings and contribution of the study at the end of discussion section.

Some abbreviations are not defined (e.g. ED, BIC-8, ULN, MACE, RCT) in the main text, some defined after their repeated usage (e.g. hs, cTn). The abbreviations defined in abstract should be defined in the main text as well.

I suggest to close the introduction section by presenting the working hypothesis.

Figures were not provided in the final form and in good resolution. Thus, it was not possible to check them properly.

It should be stated how the data are presented in the legends (SD?, SEM?)

It is not clear why p-value < 0.1 is presented in the table 1. In the method section: “A p-value below 0.05 was considered significant”. Maybe it should be 0.01?

Figure 1 is supplemental - Figure 2 can be referred to as regular Figure 1. And Figure 1 is actually Figure 3… check it please and agree numbering.

Abbreviation are not defined in the legend to the Figure 2. Make certain that all legends for the figures and tables are correct and complete.  The general recommendation is that each table and figure must stand alone.  

The Discussion part should be thoroughly revised. The authors repeated the results and statistical evaluations of the present as well as previously published studies (Biomarker-in-Cardiology 8 trial).

Author Response

We thank this reviewer for his comments. All issues were addressed.

  1. We removed ACS from the title and reworded the title.
  2. We amended the abstract to clarify some important terminology regarding randomized Groups.
  3. We added a summary statement in the abstract and a summary of the most relevant conclusion at the end of the discussion.
  4. We addressed the issue of abbreviations throughout the text. With the first appearance in text is spelled out with the abbreviation in brackets and consecutively used as abbreviation at later appearance.
  5. We added the working hypothesis at the end of the introduction.
  6. We apologize for poor reproduction Quality of figures. We resubmitted high quality resolution figures and provided figure legends with all details required. 
  7. We corrected the wrong p-value in Table 1.
  8. We apologize for wrong sequencing of figures including supplemental material alignement and amended the sequence of figures as proposed.
  9. We rephrased the discussion avoiding redundancy with results and with a better Focus on the novelty of this substudy. In addition, we added text to summarize the key findings and practical consequences. 

Round 2

Reviewer 1 Report

The authors have revised the manuscript and corrected mistakes according to the suggestion except the comparison between DMS and 0/1 or 0/3 protocol. 

Author Response

We thank the reviewer who has no further comments. Unfortunately, the design of the main Trial does not allow a comparison with the ESC 0/1 hour algorithm.

Reviewer 3 Report

The authors have answered my questions.

Author Response

We thank the reviewer who had no queries.

Reviewer 4 Report

In the abstract, the abbreviation hs-cTn is defined after its usage (as well as in the main text), acute coronary syndrome is defined twice (but not defined in the main text). What was the meaning to define MI, PCI, CABG and ULN in the abstract? I recommend revising the abbreviations in the manuscript.

Author's Note no. 5: “We added the working hypothesis at the end of the introduction.” Actually, there is no working hypothesis, only specification of the goal… and it is still not clearly presented what is the rationale for the hypothesis - “to confirm the findings of (previously published) the main BIC-8 trial after replacing all cTn values by hs-cTnT values” - this was (in the previous version) and is still poorly clarified - what is known and what is expected contribution of the proposed study.

I do not understand the reason, but figures are again of insufficient quality.

The Discussion part should be thoroughly revised. The authors repeated the results and statistical evaluations of the present as well as previously published studies (Biomarker-in-Cardiology 8 trial).

Author Response

In the abstract, the abbreviation hs-cTn is defined after its usage (as well as in the main text), acute coronary syndrome is defined twice (but not defined in the main text). What was the meaning to define MI, PCI, CABG and ULN in the abstract? I recommend revising the abbreviations in the manuscript.

We thank the reviewer and apologize for misunderstandings. We omitted the abbreviations in the abstract and started in the main text.

Author's Note no. 5: “We added the working hypothesis at the end of the introduction.” Actually, there is no working hypothesis, only specification of the goal… and it is still not clearly presented what is the rationale for the hypothesis - “to confirm the findings of (previously published) the main BIC-8 trial after replacing all cTn values by hs-cTnT values” - this was (in the previous version) and is still poorly clarified - what is known and what is expected contribution of the proposed study.

We thank the reviewer and amended the text accordingly. We introduced/rephrased the working hypothesis and added text immediately before this paragraph to clarify the rationale for this subanalysis.

I do not understand the reason, but figures are again of insufficient quality.

The insufficient image quality does not lie within our responsibility. We had submitted files containing figures in appropriate resolution.

The Discussion part should be thoroughly revised. The authors repeated the results and statistical evaluations of the present as well as previously published studies (Biomarker-in-Cardiology 8 trial).

We amended the text omitting redundancies. For this purpose, the first 7 lines were dropped and the text was expanded by almost 50% (most of the latter was carried out for R1) to discuss findings in more details.